# Increasing incidence of reported scabies infestations in the Netherlands, 2011–2021

**Babette van Deursen**[ID][1]*, **Mariëtte Hooiveld**[ID][2,3], **Susan Marks**[4], **Ingrid Snijdewind**[4], **Hans van den Kerkhof**[2], **Bas Wintermans**[5], **Ben Bom**[2], **Barbara Schimmer**[2], **Ewout Fanoy**[1]

1 Public Health Service Rotterdam-Rijnmond, Rotterdam-Rijnmond, The Netherlands, 2 National Institute for Public Health and the Environment, Bilthoven, The Netherlands, 3 Nivel, Utrecht, The Netherlands, 4 Public Health Service Haaglanden, The Hague, The Netherlands, 5 Department of Medical Microbiology, Bravis Hospital, Roosendaal, The Netherlands

* babettevandeursen@hotmail.com

## Abstract

### Introduction

Several Public Health Services and general practitioners in the Netherlands observed an increase in scabies in the Netherlands. Since individual cases of scabies are not notifiable in the Netherlands, the epidemiological situation is mostly unknown. To investigate the scabies incidence in the Netherlands, we described the epidemiology of scabies between 2011 and 2021.

### Methods

Two national data sources were analysed descriptively. One data source obtained incidence data of scabies (per 1,000 persons) of persons consulting in primary care from 2011–2020. The other data source captured the number of prescribed scabicides in the Netherlands from 2011–2021. To describe the correlation between the incidence of diagnoses and the number of dispensations between 2011 and 2020, we calculated a correlation coefficient.

### Results

The incidence of reported scabies has increased by more than threefold the last decade (2011–2020), mainly affecting adolescents and (young) adults. This was also clearly reflected in the fivefold increase in dispensations of scabicide medication during 2011–2021. The incidence and dispensations were at an all-time high in 2021. We found a strong correlation between the reported incidence and the number of dispensations between 2011 and 2020.

### Conclusions

More awareness on early diagnosis, proper treatment and treatment of close contacts is needed.

**Data Availability Statement:** All relevant data are within the paper and its Supporting Information files.

**Funding:** This study was supported by the Centre for Infectious Disease Control of the National Institute for Public Health and the Environment (RIVM), The Netherlands in the form of an unrestricted grant.

**Competing interests:** The authors have declared that no competing interests exist.

## Introduction

Scabies is officially categorized as a neglected tropical disease. In recent years scabies manifestations are increasing in numbers in Europe [1–4]. Global epidemiological data is rather scarce as it is not notifiable in many countries [5]. Scabies is mainly transmitted by prolonged skin-to-skin contact or by contact with infested items such as bedding and clothes. In the past years, the general practitioners (GPs) observed a gradual increase of scabies diagnoses in the Netherlands. Other (neighbouring) European countries also reported an increase of scabies infestations in their countries in recent years [1–4]. Individual cases of scabies in the general population are not notifiable in the Netherlands. It is only notifiable to the Public Health Service (PHS) when diagnosed in persons who works, lives or attends vulnerable settings such as kindergartens, schools, and care or cure institutions [6]. In the Netherlands, topical permethrin (5%) or topical benzyl benzoate (25%) is the preferred course of treatment for scabies. Another course of treatment is systemic by prescribing oral ivermectin (3mg) when topical treatment has failed or is contra-indicated.

Several PHS suggested anecdotally a recent increase in scabies notifications since the autumn of 2021. The number of notified outbreaks does not represent the true burden of infections in the population but are the tip of the iceberg. The epidemiological situation of scabies in the Netherlands is not well-known or even described. To investigate the scabies incidence, we analysed two national data sources to describe the epidemiology of scabies in the Netherlands between 2011 and 2021.

## Methods

Two separate national data sources were used to describe the epidemiological situation of scabies among the general population in the Netherlands during 2011–2020. To describe the correlation between the incidence of diagnoses and the number of dispensations between 2011 and 2020, we calculated a correlation coefficient (Pearson's r).

### Scabies diagnoses reported by general practitioners

The first data source contains electronic records of scabies diagnoses from a national representative primary care database of GPs hosted by Nivel (the Netherlands Institute for Health Services Research) [7]. In the Netherlands, general practitioners are the first point of contact for health care and there are no private clinics who provide primary care. We obtained incidence data of scabies (per 1,000 persons) of persons consulting general practitioners from 2011–2020 by age group and sex during the study period and performed descriptive analyses. We investigated the trend of the incidence by years and by age group. Most institutionalised residents are not taken into consideration in these registrations, considering other physicians are usually responsible for their consultations. Furthermore, the incidence of 2021 was not calculated and available at the time of our study, thus could not be taken into consideration in our analysis.

### Recorded prescriptions and over-the-counter sales of scabicides

The second database contains pharmaceutical data collected by the Dutch Foundation for Pharmaceutical Statistics (SFK) on the number of prescribed and over-the-counter scabicides in the Netherlands [8]. Data on dispensations of prescribed drugs is anonymously gathered from more than 98% of the community pharmacies in the Netherlands. We obtained the number dispensations of scabicides from SFK from 2011–2021, which consisted of type of supplied scabicide (ivermectin (3mg), permethrin (5%) and benzyl benzoate (25%)), sex and age of the patient. This data did not include any information on indication, so these dispensations could

be prescribed for other indications than scabies. We described the SFK data by sex and age group and by month. Prescriptions of institutionalised residents are not gathered by SFK.

## Ethical statement

This study used routinely collected, anonymized and aggregated data, which cannot be traced back to individual patients. Surveillance of infectious diseases is one of the legal tasks of the Public Health Service as described under the Public Health Act, and do not require separate medical ethical clearance.

## Results

### Reported scabies diagnoses

The incidence of scabies diagnoses per 1,000 persons per year recorded by GPs in 2011 was 0.6 and increased more than 4-fold to 2.6 in 2020 (Fig 1). No differences were observed between men and women. The highest incidence was among 15–29-year-old persons, especially among 20–24-year-olds. The increase started in 2013 and continued the following years. There was a stabilisation in incidence in 2018 yet increased again in 2020 to 2.6 scabies diagnoses per 1,000 persons (per year).

### Recorded scabicides prescriptions and over-the counter sales by public pharmacies

A total of 702,317 scabicides (permethrin, ivermectin and benzyl benzoate) were dispensed between 2011 and 2021. In 14% of the dispensations, sex of the patient was not registered. Slightly more males bought a scabicide then women (53% vs. 47%) and the male-to-female ratio remained stable during the study period. The most common prescribed scabicide was permethrin (>70%).

In 2011, the total number of dispensations of all scabicides was 28,300 and increased to a total of 142,622 in 2021, which is more than a 5-fold increase in a decade. In more detail: since the end of 2013, the number increased gradually until 2018 and became more pronounced

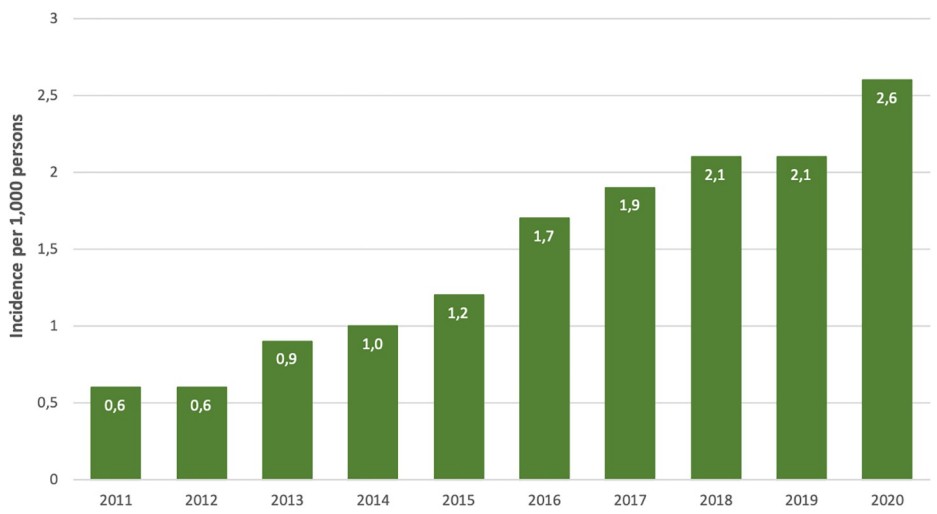

**Fig 1. The incidence of scabies diagnoses per 1,000 persons as recorded by the general physicians from 2011–2020, the Netherlands.**

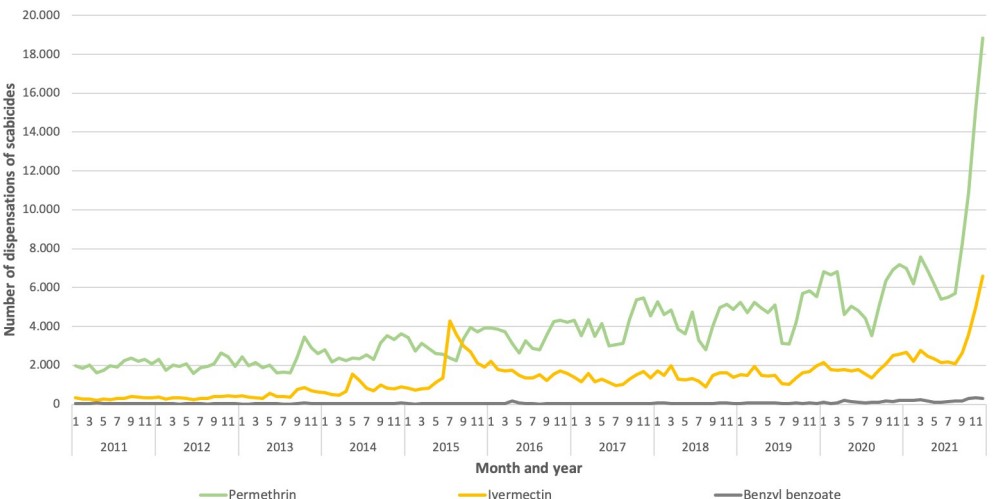

**Fig 2. Number of dispensations of scabicides (permethrin, ivermectin and benzyl benzoate) per month between 2011–2021, the Netherlands.**

during 2019 and especially peaked in 2021 (Fig 2). In the last months of 2021, the number of permethrin dispensations even reached an all-time high with an increase of >200% from September–December. Furthermore, the increase in dispensations of permethrin indicates a seasonal pattern with a clear peak in autumn and winter months. We found a strong correlation between the number of dispensations and reported incidence between 2011 and 2020 (r = 0.98; p<0.001).

Most of the permethrin and ivermectin dispensations were handed out to patients in the age category of 20–25 years (19%), followed by 15–20 years (9%) and 25–30 years (8%) (Fig 3). Approximately 70% of the dispensations was permethrin in each age group, except for 0–15-year-olds where it was around 80%. The trend of dispensations by age group was similar to the trend seen in Fig 2. The 'unknown' group (18%) among permethrin dispensations are

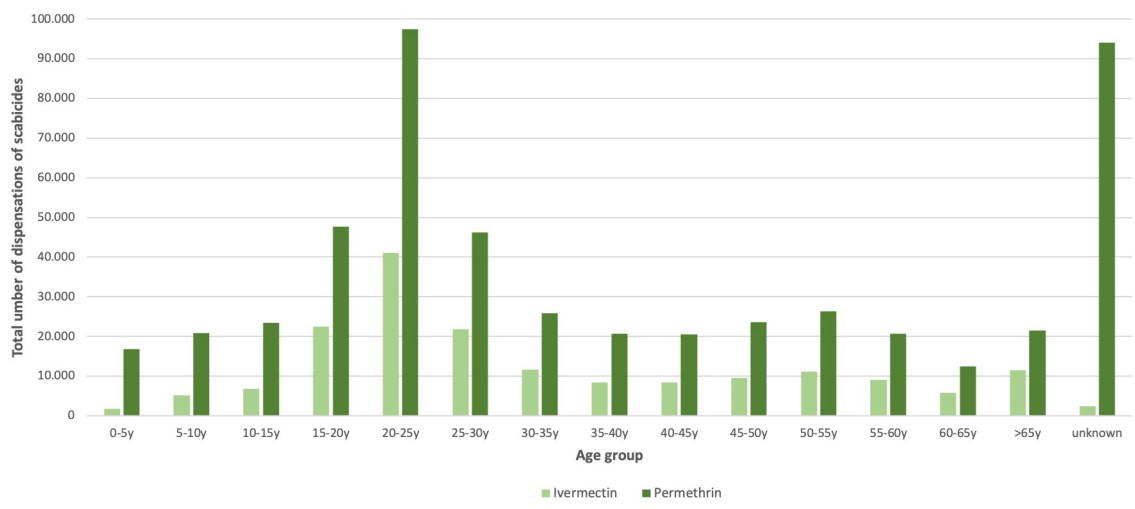

**Fig 3. Total number of dispensations of permethrin and ivermectin by age group, 2011–2021, the Netherlands.**

most likely over-the-counter sales, since no prescription for this drug is required. For ivermectin, the 'unknown' dispensations are most likely from doctors/general practitioners to other colleagues or to use as stock.

## Discussion

The incidence of reported scabies in the general population of the Netherlands has increased by more than threefold the last decade. Incidence of diagnoses made by general practitioners started to increase gradually since 2013 and continued the following years, especially among adolescents and (young) adults. The number of dispensations of scabicides reported a more than fivefold increase from 2011–2021. The reported incidence of scabies and the number of dispensations of scabicides between 2011 and 2020 were strongly correlated. Most of the dispensations were permethrin, mainly dispensed to 20–25-year-olds and mostly during autumn and winter season. The observed peak in ivermectin prescriptions in 2014–2015 was due to several large outbreaks in healthcare facilities with a high number of contacts preventively treated [9].

It is interesting to assess the impact of the COVID-19 pandemic on scabies incidence, since almost all notifiable infectious diseases decreased during the COVID-19 pandemic due to less transmission as underreporting and underdiagnosing [10]. On the contrary, there was a remarkable sharp increase of incidence and dispensations of scabicides since 2020, mainly in adolescents and (young) adults. There seems no limiting effect of the general social distancing measures on scabies incidence. It is suggested that some of the measures had the opposite effect among the adolescents and young adults. For example, a curfew between 23 January 2021 and 28 April 2021 was in place in the Netherlands, which could have led to more sleepovers and thus more transmission. The recent trends of scabies incidence in other European countries during the COVID-19 pandemic is unclear, as no peer-reviewed studies have been published beyond the year 2019.

The observed increase in incidence corresponds with the trend in other European countries, such as Germany, Norway, Croatia, Poland, and Spain [1–4,11]. Although each country has slightly different surveillance systems in place or used other data sources than our study, they all still reported an increase in the last years. The increase in number of dispensations of scabicides could partly explained by population growth in recent years. The increasing trends were most striking in adolescents and (young) adults and was also timed roughly similarly. It is remarkable that these countries observed a comparable seasonal trend in autumn and winter season [1–4,11,12]. It can be suggested that the seasonal trend is caused by more contact during cold and dry seasons [12].

Since skin-to-skin contact is the common transmission route for scabies, the observed increase among adolescents and (young) adults could be explained by more exposure due to more contacts than older generations [1,4]. We did not see a difference in incidence between males and females, however more males bought a scabicide medication than females. In Germany, more males were diagnosed with scabies in recent years and it was suggested that young adult males have the highest social connectivity [4]. It is also known that, concerning scabies, medical and outbreak policy adherence (such as washing of clothing and bed linen) among adolescents and young adults are generally lower than for older adults, which can lead to treatment failure [13]. Other studies suggest that the increase could be partly explained by frequent introductions from tourists or asylum seekers [2].

Increasing resistance of mites to scabicides could be an explanation on the increasing trend [14]. Unfortunately, we do not have insight in prevalence of scabicide resistance as diagnostic

tools are not yet in place. Further research on resistance, for example by developing diagnostic molecular tools, is necessary to fully understand the drivers of the increasing trend.

This study was based on persons who consulted their GP and/or who bought scabicide medication at the pharmacy, which means that institutionalised residents are not included. However, most of the infested persons live and work outside these facilities. The reported incidence of 2021 was not calculated and still unknown, therefore we could not take it into consideration in our study. Also, we did not investigate the number of reported outbreaks to the PHS because the criteria for notifying scabies in vulnerable settings were adapted in 2019 and therefore not comparable with recent years. It is possible that the increase in dispensations is caused by patients with multiple prescriptions due to treatment failure. However, we only obtained data on the total number of dispensations of scabicides and not number of patients, so we could not investigate if the increase is explained by treatment failure, re-infestations or by more patients with scabies infestations. Additionally, we do not obtain the data of the other 2% of community pharmacies since they are not included in the SFK dataset. Also, permethrin can be bought at drugstore chains in the Netherlands, so the number of all dispensations is presumably higher than presented. Another limitation is that SFK does not report the indication for the prescription or dispensation, so it is possible that some of the dispensations were not related to scabies. For example, ivermectin could also be prescribed as a drug against COVID-19, while it is proven not to be effective [15,16]. This could explain the sudden rise of ivermectin dispensations at the end of 2021.

The surveillance of scabies is scattered, and an integration of data sources would be wise to improve since it is not notifiable for individual cases. Some elements are missing, such as social background of patients, source of infestation and transmission routes of mites within and between countries. Skin material scraped from suspect lesions or bed linen can be used as diagnostic tool using PCR [17]. Positive specimens can be molecularly typed and combined with epidemiological information, which can generate insight in transmission chains within the population and identify circulation within social communities or facilities. These deeper understandings of hotspots and transmission routes are informative for effective anti-scabies measures and policy.

In summary, the incidence of scabies and the dispensations of scabicide medications in the Netherlands have increased in the last decade, especially among adolescents and (young) adults. As this scabies increase is observed in several more European countries, a robust integrated approach is needed. The surveillance of scabies should be improved by combining several available databases and by collecting more data on social background and source of infestation. The potential use of molecular typing to detect clusters and transmission chains should be explored. Higher awareness of suspected clinical symptoms among the younger population and physicians is necessary to allow early diagnosis and treatment and thus stop transmission, so therefore we recommend better training and clear medical guidelines for physicians. In addition to these advices, it is important to emphasize proper treatment including washing of clothing and bed linen. Furthermore, the treatment should not only be focused on the individual patient, but also on close contacts.

## Supporting information

**S1 Database.**
(DOCX)

**S1 File.**
(XLSX)

## Acknowledgments

The authors would like to thank other colleagues of the Public Health Service and National Institute for Public Health and the Environment for their input and advice. This study has been approved according to the governance code of Nivel Primary Care Database, under number NZR-00321.072.

## Author Contributions

**Conceptualization:** Susan Marks, Ingrid Snijdewind, Ewout Fanoy.

**Data curation:** Mariëtte Hooiveld, Ben Bom.

**Formal analysis:** Babette van Deursen.

**Investigation:** Babette van Deursen, Susan Marks, Ingrid Snijdewind, Ewout Fanoy.

**Methodology:** Babette van Deursen, Mariëtte Hooiveld, Ben Bom.

**Supervision:** Ewout Fanoy.

**Validation:** Mariëtte Hooiveld, Hans van den Kerkhof, Bas Wintermans, Ben Bom, Barbara Schimmer, Ewout Fanoy.

**Visualization:** Babette van Deursen, Ben Bom, Ewout Fanoy.

**Writing – original draft:** Babette van Deursen.

**Writing – review & editing:** Babette van Deursen, Mariëtte Hooiveld, Susan Marks, Ingrid Snijdewind, Hans van den Kerkhof, Bas Wintermans, Ben Bom, Barbara Schimmer, Ewout Fanoy.

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
