## [Decision Letter · Decision Letter 0]

11 Mar 2022

PONE-D-22-04888Increasing incidence of scabies infestations in the Netherlands, 2011 – 2021PLOS ONE

Dear Dr. van Deursen,

Thank you for submitting your manuscript to PLOS ONE. After careful consideration, we feel that it has merit but does not fully meet PLOS ONE’s publication criteria as it currently stands. Therefore, we invite you to submit a revised version of the manuscript that addresses the points raised during the review process.

Both reviewers agree that the paper has its merits but needs some additional work. Please address all of their comments before resubmitting.

We look forward to receiving your revised manuscript.

Kind regards,

Joël Mossong

Academic Editor

PLOS ONE

Journal Requirements:

Reviewers' comments:

Reviewer's Responses to Questions

**Comments to the Author**

1. Is the manuscript technically sound, and do the data support the conclusions?

Reviewer #1: Partly

Reviewer #2: Yes

2. Has the statistical analysis been performed appropriately and rigorously? 

Reviewer #1: No

Reviewer #2: N/A

3. Have the authors made all data underlying the findings in their manuscript fully available?

Reviewer #1: No

Reviewer #2: No

4. Is the manuscript presented in an intelligible fashion and written in standard English?

Reviewer #1: Yes

Reviewer #2: Yes

5. Review Comments to the Author

Reviewer #1: General comment

Overall, even if the topic is quite relevant, the paper is quite poor. The intro and methods sections need to be improven and completed.

Abstract

The main objective is too ambitious, a research shouldn´t be triggered by an assumption, please rephrase

Introduction

Overall, the introduction is quite short. I would recommend to add a short description on disease background, not only on clinics and current challenges (i.e. treatment resistances) but also on its worldwide epidemiology (including the fact that it´s an NTD)

Line 36: Is it possible to sustain this information (“general practitioners (GPs) observed a gradual increase of scabies diagnoses in 37 the Netherlands”) with some references?

Line 40: Are the PHS notified of individuals attending this kind of vulnerable setting? Or are you referring to aggregate/outbreak data? Please specify

Line 46: same comment that above. The main objective is too ambitious, a research question cannot be based in previous assumptions

Methods

Line 55: Would it be possible to include some information of population coverage?

Is there any other available data source to understand the situation on institutionalized residents? Why the authors did not analyze the information notified to the PHS on vulnerable settings?

Which was the study period for this first data source? Does it gather info also from private clinics?

Line 58. Please describe in detail which sort of descriptive analysis was performed. Results by sex are also given in the results section, but this is not explained in methods.

Line 67. Are these treatments only used for scabies? Please specify

Further spatial analysis could have been performed based on the pharma data

Ethical statement: There is a reference to outbreak investigations, but this information is not included in the analysis, is this correct?

Results

This section is a bit poor. A table with some additional data is recommended.

Was the study period the same? Did you find any temporal relationship between both databases?

Line 87. This last sentence should be placed in the discussion section, as it´s not referring to study results.

A bit more advanced analytical techniques (such as linear regression, joint point analysis) could help to better assess the temporal trend of this disease

Line 105-108: this belong to the discussion section

Discussion

Line 124. This first statement cannot be based on the provided results (due to the limitations of both data sources)

Line 131 It is probable that the incidence of notifiable diseases has decreased during the COVID-pandemic, but also there might have been underreporting. Regarding scabies, and given its transmission mode (which should be better explained in the introduction section), the impact of the COVID pandemic might be the opposite (somehow, homes became close institutions). Also, the high burden on GPs might have also affect the attendance and follow-up of patients, worsen the situation.

Line 139. What is the relationship between population growth and increase in scabies incidence? Is this related to worst health conditions and poverty?

Regarding the increasing trend in young population, is it possible that sexual transmission of scabies has also increased?

Line 154. I don´t think this is an explanation of COVID times, on the contrary, incidence should have decreased.

Line 157. This paragraph does not discuss the study results. Please rephrase or delete

Line 163. Even if you don´t have info on resistance, an increase in other treatments rather than permethrin could had supported this hypothesis.

Line 180. You may add a reference to EMA recommendations against the use of ivermectin for COVID-19 EMA advises against use of ivermectin for the prevention or treatment of COVID-19 outside randomised clinical trials | European Medicines Agency (europa.eu)

Line 197. Looking at the incresaing number of cases, would you also recommend better training, medical guidelines, etc for primary health care centres?

Reviewer #2: The authors present an increasing incidence of scabies infestations in the Netherlands during 2011 – 2021. The manuscript confirms what previously reported by other countries in Europe, as well as providing additional information during more recent years under the pandemic context. The manuscript is interesting and deserve to be published, here below some points that could improve clarity and the quality of the manuscript.

Language revision, preferably by a native speaker, would improve the quality of the manuscript. For example, the use of the word ‘several’ would fit better than ‘multiple’ in some sentences. In the title ‘scabies diagnoses by general practitioners’, I would add the word ‘reported’ before ‘by’. Similarly for the title, ‘Scabicides prescriptions and over-the counter sales by public pharmacies’, I would add ‘reported’ or ‘recorded’. Line 150, I would replace ‘since’ with ‘in’

Introduction, line 44, I think the sentence ‘There are complexities concerning scabies surveillance: clinical recognition and diagnostic tools such as microscopy, dermatoscopy and PCR have sensitivity issues’ is a bit out of place since it is not a notifiable disease in the Netherlands, and does not connect well the sentences before and after. It could be deleted from here and be used in the discussion part when relevant.

Methods part, line 52, it is mentioned ‘2011-2020’ while in the introduction and abstract it is mentioned 2011-2021 period. This is confusing to the reader. My understanding is that the data on scabies diagnosis was available until 2020. If this is the case, authors should clarify somewhere in the manuscript the reason why data from 2021 were not included.

Methods part, line 68, could you specify more in details the type of treatment? E.g., permethrin cream? Ivermectin tablets? Is It possible to add in introduction which treatments are used in the Netherlands?

Ethical statement, line 73, the authors stated, ‘outbreak investigations of notifiable diseases such as scabies’ while in line 38 ‘cases of scabies are not notifiable’. I would suggest rephrasing to make it more understandable what you mean in the ‘ethical statement’ section.

Results, line 87, ‘Remarkably, this was despite the COVID-19 pandemic and its corresponding control and hygiene measures.’ I think it is a very interesting results, do you have any hypothesis on this aspect? Could you add a sentence or short paragraph in discussion section providing an explanation/hypothesis of this phenomenon based on your experience?

Results, line 83, ‘No differences were observed between men and women.’ while line 97 ‘Slightly more males bought a scabicide then women’. Is there a possible explanation for this difference?

Do data on scabies diagnosis include recurrent cases (re-infestation) or it was possible to identify records from the same patient in your dataset? This need to be clarified and added as limitation if was not possible to look at re-occurrent cases.

It is mentioned in line 59 ‘Most institutionalised residents are not taken into consideration in these registrations, considering other physicians are usually responsible for their consultations.’ What about the database on pharmaceutical data, does it exclude as well the treatment of institutionalized residents? If not, this need to be clarified and/or added as limitation.

Line 180 ‘ivermectin could also be prescribed as a drug against COVID-19, while it is proven not to be effective (15). This can explain the sudden rise of ivermectin dispensations at the end of 2021.’ Including data on scabies diagnosis in 2021 would allow to have more solid conclusion. If you did not compare it with scabies diagnosis data, I would amend ‘this can explain’ with ‘this could explain’.

Line 183 and line 193, the authors mention ‘surveillance’ but it is not a notifiable disease, could you clarify and rephrase it if needed?

Figure 1: Did you try to compare the incidences level reported in the Netherlands with the ones reported in other countries? Anything to highlight? A sentence/paragraph on this aspect can be added in the discussion providing authors’ perspective.

Figure 2: does the 3-months average add something to the interpretation of results or discussion. If not, I would simplify it deleting them for permethrin and ivermectin.

I would also integrate figure 1 with figure 2, visually comparing the two datasets on scabies diagnosis and dispensations of scabicides. You could also use number of diagnoses instead of incidence, if this will improve the data visualization. The values on incidence per year are available in supplementary material in any case.

Figure 3: Do you see any difference in terms of percentage between ivermectin and permethrin by age group? If so, you could also describe and discuss it.

I think would be also nice to add an additional figure showing the trend of incidence per age-group during the study period using the data reported in table 2, supplementary material.

6. PLOS authors have the option to publish the peer review history of their article (what does this mean?). If published, this will include your full peer review and any attached files.

Reviewer #1: **Yes: **Zaida Herrador

Reviewer #2: No

---

## [Author Response · Author response to Decision Letter 0]

25 Apr 2022

Response to reviewers

We would like to thank the reviewers for their time and comments on our manuscript. They were really helpful and valuable for improving our manuscript. Below we give a point-by-point response to the reviewers. 

Reviewer #1: 

General comment

Overall, even if the topic is quite relevant, the paper is quite poor. The intro and methods sections need to be improven and completed.

Abstract

The main objective is too ambitious, a research shouldn´t be triggered by an assumption, please rephrase. We adjusted the aim of the study, see line 20 and 59.

Introduction

Overall, the introduction is quite short. I would recommend to add a short description on disease background, not only on clinics and current challenges (i.e. treatment resistances) but also on its worldwide epidemiology (including the fact that it´s an NTD). Great suggestion, we added some additional information on disease background, epidemiology and treatment. See lines 40 and 49.

Line 36: Is it possible to sustain this information (“general practitioners (GPs) observed a gradual increase of scabies diagnoses in 37 the Netherlands”) with some references? It is partly based on the presented incidence number in our study; however, it was never published in a manuscript before. So therefore, we cannot sustain it with references. 

Line 40: Are the PHS notified of individuals attending this kind of vulnerable setting? Or are you referring to aggregate/outbreak data? Please specify. We clarified in the text, see line 48.

Line 46: same comment that above. The main objective is too ambitious, a research question cannot be based in previous assumptions. We adjusted the aim of the study, see line 59.

Methods

Line 55: Would it be possible to include some information of population coverage?

Is there any other available data source to understand the situation on institutionalized residents? Why the authors did not analyze the information notified to the PHS on vulnerable settings? We clarified in the discussion, see line 208.

Which was the study period for this first data source? Does it gather info also from private clinics? We clarified in the text, see line 72and 74.

Line 58. Please describe in detail which sort of descriptive analysis was performed. Results by sex are also given in the results section, but this is not explained in methods. We clarified in the text, see lines 75 and 88.

Line 67. Are these treatments only used for scabies? Please specify. We clarified in the text, see lines 87.

Further spatial analysis could have been performed based on the pharma data. We don’t have data on the location of the dispensations, so therefore spatial analysis could not have been performed. 

Ethical statement: There is a reference to outbreak investigations, but this information is not included in the analysis, is this correct? That is correct, thank you for noticing. We adjusted our ethical statement, see lines 95.

Results

This section is a bit poor. A table with some additional data is recommended. Was the study period the same? Did you find any temporal relationship between both databases? A bit more advanced analytical techniques (such as linear regression, joint point analysis) could help to better assess the temporal trend of this disease. Great suggestion, we calculated a correlation coefficient to describe the correlation between the incidence of diagnoses and the number of dispensations between 2011 and 2020, see lines 65 and 130.

Line 87. This last sentence should be placed in the discussion section, as it´s not referring to study results. Thank you, we removed it from results section.

Line 105-108: this belong to the discussion section. We removed it to discussion section, see lines 157 – 159.

Discussion

Line 124. This first statement cannot be based on the provided results (due to the limitations of both data sources). We added the word ‘ reported’, since is it not an notifiable disease and thus dependent on patients who consult the GPs, see line 151.

Line 131. It is probable that the incidence of notifiable diseases has decreased during the COVID-pandemic, but also there might have been underreporting. Regarding scabies, and given its transmission mode (which should be better explained in the introduction section), the impact of the COVID pandemic might be the opposite (somehow, homes became close institutions). Also, the high burden on GPs might have also affect the attendance and follow-up of patients, worsen the situation. Great suggestion, we’ve added it to the discussion, see line 162.

Line 139. What is the relationship between population growth and increase in scabies incidence? Is this related to worst health conditions and poverty? Thank you for noticing, it was in the wrong place and is now replaced in line 176.

Regarding the increasing trend in young population, is it possible that sexual transmission of scabies has also increased? Great suggestion, it is possible since it is transmissible by prolonged skin-to-skin contact. We mention sleepovers as well in our discussion, see line 165.

Line 154. I don´t think this is an explanation of COVID times, on the contrary, incidence should have decreased. We rephrased the paragraph, see lines 182 -192.

Line 157. This paragraph does not discuss the study results. Please rephrase or delete. We integrated with another paragraph in our discussion, see lines 223 – 233.

Line 163. Even if you don´t have info on resistance, an increase in other treatments rather than permethrin could had supported this hypothesis. Interesting suggestion, however as we don’t have any information on re-infestations, failed treatment and indication for the prescription, we cannot claim that the increase in other scabicides can be explained by resistance. 

Line 180. You may add a reference to EMA recommendations against the use of ivermectin for COVID-19 EMA advises against use of ivermectin for the prevention or treatment of COVID-19 outside randomised clinical trials | European Medicines Agency (europa.eu). Thank you for the reference, we included it in the discussion.

Line 197. Looking at the incresaing number of cases, would you also recommend better training, medical guidelines, etc for primary health care centres? Great suggestion, we added it in line 242.

 

Reviewer #2:

The authors present an increasing incidence of scabies infestations in the Netherlands during 2011 – 2021. The manuscript confirms what previously reported by other countries in Europe, as well as providing additional information during more recent years under the pandemic context. The manuscript is interesting and deserve to be published, here below some points that could improve clarity and the quality of the manuscript.

Language revision, preferably by a native speaker, would improve the quality of the manuscript. For example, the use of the word ‘several’ would fit better than ‘multiple’ in some sentences. In the title ‘scabies diagnoses by general practitioners’, I would add the word ‘reported’ before ‘by’. Similarly for the title, ‘Scabicides prescriptions and over-the counter sales by public pharmacies’, I would add ‘reported’ or ‘recorded’. Line 150, I would replace ‘since’ with ‘in’. Thank you for your suggestions, we revised our manuscript following your comments.

Introduction, line 44, I think the sentence ‘There are complexities concerning scabies surveillance: clinical recognition and diagnostic tools such as microscopy, dermatoscopy and PCR have sensitivity issues’ is a bit out of place since it is not a notifiable disease in the Netherlands, and does not connect well the sentences before and after. It could be deleted from here and be used in the discussion part when relevant. We removed the sentence, it will be discussed in the discussion, see lines 223 – 233.

Methods part, line 52, it is mentioned ‘2011-2020’ while in the introduction and abstract it is mentioned 2011-2021 period. This is confusing to the reader. My understanding is that the data on scabies diagnosis was available until 2020. If this is the case, authors should clarify somewhere in the manuscript the reason why data from 2021 were not included. This is correct, thank you for highlighting. We added some information on why we could not include the incidence of 2021 in our study, see lines 78 and 207.

Methods part, line 68, could you specify more in details the type of treatment? E.g., permethrin cream? Ivermectin tablets? Is It possible to add in introduction which treatments are used in the Netherlands? We added some information on treatment in the introduction, see line 49.

Ethical statement, line 73, the authors stated, ‘outbreak investigations of notifiable diseases such as scabies’ while in line 38 ‘cases of scabies are not notifiable’. I would suggest rephrasing to make it more understandable what you mean in the ‘ethical statement’ section. Thank you for noticing, we adjusted the ethical statement, see line 95.

Results, line 87, ‘Remarkably, this was despite the COVID-19 pandemic and its corresponding control and hygiene measures.’ I think it is a very interesting results, do you have any hypothesis on this aspect? Could you add a sentence or short paragraph in discussion section providing an explanation/hypothesis of this phenomenon based on your experience? We have included our hypothesis in the discussion, see lines 165 – 168.

Results, line 83, ‘No differences were observed between men and women.’ while line 97 ‘Slightly more males bought a scabicide then women’. Is there a possible explanation for this difference? Great question. We don’t have a distinct explanation for this difference, however, Germany observed that more males were diagnosed with scabies. We added this in the discussion, see lines 185 -188.

Do data on scabies diagnosis include recurrent cases (re-infestation) or it was possible to identify records from the same patient in your dataset? This need to be clarified and added as limitation if was not possible to look at re-occurrent cases. This is mentioned as a limitation in the discussion, see lines 211-215.

It is mentioned in line 59 ‘Most institutionalised residents are not taken into consideration in these registrations, considering other physicians are usually responsible for their consultations.’ What about the database on pharmaceutical data, does it exclude as well the treatment of institutionalized residents? If not, this need to be clarified and/or added as limitation. We clarified in the text, see lines 205 and 206.

Line 180 ‘ivermectin could also be prescribed as a drug against COVID-19, while it is proven not to be effective (15). This can explain the sudden rise of ivermectin dispensations at the end of 2021.’ Including data on scabies diagnosis in 2021 would allow to have more solid conclusion. If you did not compare it with scabies diagnosis data, I would amend ‘this can explain’ with ‘this could explain’. We rephrased the text, see lines 221.

Line 183 and line 193, the authors mention ‘surveillance’ but it is not a notifiable disease, could you clarify and rephrase it if needed? We rephrased the text, see lines 224 and 238.

Figure 1: Did you try to compare the incidences level reported in the Netherlands with the ones reported in other countries? Anything to highlight? A sentence/paragraph on this aspect can be added in the discussion providing authors’ perspective. We compare the trends in incidence and in dispensations of other countries in the discussion, see lines 172-180. 

Figure 2: does the 3-months average add something to the interpretation of results or discussion. If not, I would simplify it deleting them for permethrin and ivermectin. Thank you for your suggestion, we removed the 3-moths average, see the revised figure 2. 

I would also integrate figure 1 with figure 2, visually comparing the two datasets on scabies diagnosis and dispensations of scabicides. You could also use number of diagnoses instead of incidence, if this will improve the data visualization. The values on incidence per year are available in supplementary material in any case. 

Great suggestion, however, the incidence data is only available per year which would be hard to read if you overlap both figures. The pharmaceutical data is relevant by month because of the seasonality. Furthermore, we now calculate the correlation coefficient to describe the correlation between the incidence of diagnoses and the number of dispensations between 2011 and 2020. Therefore, we decided not to integrate figure 1 and 2. 

Figure 3: Do you see any difference in terms of percentage between ivermectin and permethrin by age group? If so, you could also describe and discuss it. We included it in the results section, see line 138.

I think would be also nice to add an additional figure showing the trend of incidence per age-group during the study period using the data reported in table 2, supplementary material. Great suggestion, however, when the number of dispensations by age group per month is plotted, it gives the same trend as seen in Figure 2 which will not give any added value visually. We mention it in the results, see line 139.

---

## [Decision Letter · Decision Letter 1]

10 May 2022

Increasing incidence of reported scabies infestations in the Netherlands, 2011 – 2021

PONE-D-22-04888R1

Dear Dr. van Deursen,

We’re pleased to inform you that your manuscript has been judged scientifically suitable for publication and will be formally accepted for publication once it meets all outstanding technical requirements.

Kind regards,

Joël Mossong

Academic Editor

PLOS ONE

Additional Editor Comments (optional):

Reviewers' comments:

Reviewer's Responses to Questions

**Comments to the Author**

1. If the authors have adequately addressed your comments raised in a previous round of review and you feel that this manuscript is now acceptable for publication, you may indicate that here to bypass the “Comments to the Author” section, enter your conflict of interest statement in the “Confidential to Editor” section, and submit your "Accept" recommendation.

Reviewer #2: All comments have been addressed

2. Is the manuscript technically sound, and do the data support the conclusions?

Reviewer #2: (No Response)

3. Has the statistical analysis been performed appropriately and rigorously? 

Reviewer #2: (No Response)

4. Have the authors made all data underlying the findings in their manuscript fully available?

Reviewer #2: (No Response)

5. Is the manuscript presented in an intelligible fashion and written in standard English?

Reviewer #2: (No Response)

6. Review Comments to the Author

Reviewer #2: (No Response)

7. PLOS authors have the option to publish the peer review history of their article (what does this mean?). If published, this will include your full peer review and any attached files.

Reviewer #2: No

---

## [Editor Report · Acceptance letter]

6 Jun 2022

PONE-D-22-04888R1 

Increasing incidence of reported scabies infestations in the Netherlands, 2011 – 2021 

Dear Dr. van Deursen:

I'm pleased to inform you that your manuscript has been deemed suitable for publication in PLOS ONE. Congratulations! Your manuscript is now with our production department. 

Kind regards, 

on behalf of

Dr. Joël Mossong 

Academic Editor

PLOS ONE